# Detached and Interactive Multimodal Learning

## ABSTRACT

Recently, Multimodal Learning (MML) has gained significant interest as it compensates for single-modality limitations through comprehensive complementary information within multimodal data. However, traditional MML methods generally use the *joint* learning framework with a uniform learning objective that can lead to the modality competition issue, where feedback predominantly comes from certain modalities, limiting the full potential of others. In response to this challenge, this paper introduces DI-MML, a novel *detached* MML framework designed to learn complementary information across modalities under the premise of avoiding modality competition. Specifically, DI-MML addresses competition by separately training each modality encoder with isolated learning objectives. It further encourages cross-modal *interaction* via a shared classifier that defines a common feature space and employing a dimension-decoupled unidirectional contrastive (DUC) loss to facilitate modality-level knowledge transfer. Additionally, to account for varying reliability in sample pairs, we devise a certainty-aware logit weighting strategy to effectively leverage complementary information at the instance level during inference. Extensive experiments conducted on audio-visual, flow-image, and front-rear view datasets show the superior performance of our proposed method.

## CCS CONCEPTS

• **Do Not Use This Code → Generate the Correct Terms for Your Paper**; *Generate the Correct Terms for Your Paper*; Generate the Correct Terms for Your Paper; Generate the Correct Terms for Your Paper.

## KEYWORDS

Multimodal Learning, Modality Competition, Cross-modal Interaction, Dimension-decoupled Unidirectional Contrastive Loss

**ACM Reference Format:**
Anonymous Author(s). 2024. Detached and Interactive Multimodal Learning. In *Proceedings of Proceedings of the 32th ACM International Conference on Multimedia (MM '24).* ACM, New York, NY, USA, 10 pages. https://doi.org/XXXXXXX.XXXXXXX

## 1 INTRODUCTION

Multimodal learning (MML) has emerged to enable machines to better perceive and understand the world with various types of data, which has already been applied to autonomous driving [35], sentiment analysis [17], medical health [1], etc. Data from different

modalities may contain distinctive and complementary knowledge, which allows MML outperforms unimodal learning [14]. Despite the advances in MML, fully exploiting the information from multimodal data still remains challenging.

Recent studies [15, 30] have found that the unimodal encoder in MML underperforms its best unimodal counterpart trained independently. Huang et al. [15] attribute the cause of this phenomenon to *modality competition*, where the dominant modality hinders the learning of other weak modalities, resulting in imbalanced modality-wise performance. Existing solutions [9, 22, 41] mainly try to modulate and balance the learning paces of different modalities, which generally follow the joint training framework and a uniform learning objective is employed for all modalities, as shown in Figure 1. However, according to [8], the fused uniform learning objective is actually the reason for modality competition since the backward gradient predominantly comes from certain better modalities, hindering the learning of others, as illustrated in Figure 2. Meanwhile, [6] has declared that despite the competition between modalities, the interactions in joint training can facilitate the exploitation of multimodal knowledge. Therefore, existing solutions are caught in the dilemma of mitigating competition and facilitating interactions, where the competition issue has not been eradicated, limiting further improvements in multimodal performance.

In this paper, we empirically reveal that eliminating modality competition may be more critical for multimodal learning, which motivates us to design a competition-free training scheme for MML. Therefore, we decide to abandon the joint training framework and construct the *detached* learning process via assigning each modality with isolated learning objectives. Although the naive detached framework, i.e., performing unimodal training independently, could avoid modality competition, it still suffers from the following two challenges, limiting its further improvement.

- **Disparate feature spaces**. The intrinsic heterogeneity between modalities usually requires different processing strategies as well as model structures, which may lead to disparate feature spaces based on independent unimodal training and then pose a great challenge on fusing the extracted multimodal knowledge.
- **Lack of cross-modal interactions.** The cross-modal interactions can help to facilitate the exploitation of multimodal knowledge. However, independent unimodal training insulates the interactions for both encoder training and multimodal prediction process, limiting the learning and exploitation of multimodal complementary information.

To address all above issues, we propose a novel DI-MML that achieves cross-modal **I**nteractions under the **D**etached training scheme. Unlike independent unimodal training, we first apply an additional shared classifier to regulate a shared feature space for various modalities, alleviating the difficulty on fusion process. To encourage cross-modal interactions during encoder training, we propose a Dimension-decoupled Unidirectional Contrastive (DUC) loss to transfer the modality-level complementary knowledge. We

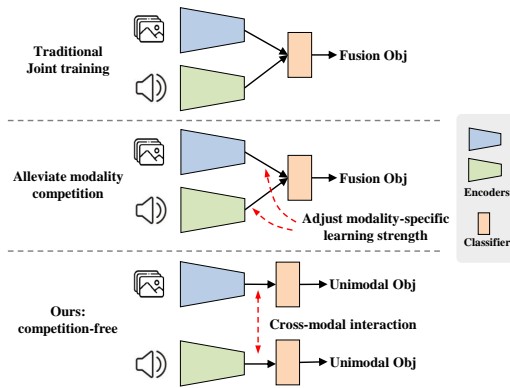

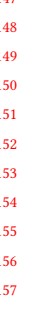

Figure 1: The difference between previous methods with ours. Only our method abandons the uniform fusion objective and updates each modal network with isolated objectives.

introduce the dimension-wise prediction to evaluate the discriminative knowledge for each dimension and then divide feature dimensions into effective and ineffective groups, enabling the complementary knowledge transfer within modalities and maintaining the full learning of each modality itself. Further, to enhance interactions during multimodal prediction, we then freeze the learned encoders and train a fusion module. Considering that there may be reliability disparities between modalities in sample pairs, we devise a certainty-aware logit weighting strategy during inference so that we can fully utilize the complementarities at the instance level.

Our main contributions can be summarized as follows:

- To the best of our knowledge, this paper is the first to completely avoid modality competition while ensuring complementary cross-modal interactions in MML. We propose a novel DI-MML framework that trains each modality with isolated learning objectives.
- We design a shared classifier to regulate a shared feature space and a Dimension-decoupled Unidirectional Contrastive (DUC) loss to enable sufficient cross-modal interactions, which exploits modality-level complementarities.
- During inference, we utilize the instance-level complementarities via a certainty-aware logit weighting strategy.
- We perform extensive experiments on four datasets with different modality combinations to validate superiority of DI-MML and its effectiveness on competition elimination.

## 2 RELATED WORK

### 2.1 Modality Competition in MML

Multimodal learning is expected to outperform the unimodal learning scheme since multiple signals generally bring more information [14]. However, recent research [30] has observed that the multimodal joint training network underperforms the best unimodal counterpart. Besides, even if the multimodal network surpasses the performance of the unimodal network, the unimodal encoders from multimodal joint training perform worse than those from unimodal training [5, 32, 33]. This phenomenon is termed as "modality competition" [15], which suggests that each modality cannot be fully learned especially for weak modalities since there exists inhibition

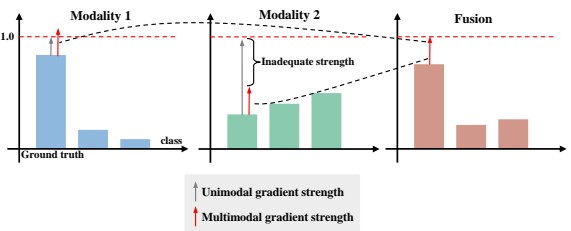

Figure 2: Modality competition comes from uniform learning objective. The columns represent predicted probabilities for each class. The fused prediction is dominated by modality 1 (better), resulting in a significant gap between the fusion gradient and the gradient needed for modality 2 (weak).

between them. Researchers have proposed various methods to address this challenge, including gradient modulation [8, 22], learning rate adjustment [27, 41], knowledge distillation [6], etc. Despite their improvement, the competition phenomenon still exists since they insist on leveraging joint training scheme with a uniform learning objective, which is the culprit for modality competition [8]. The preserved competition greatly limits the improvement of multimodal performance. In this paper, we aims to design a competition-free MML scheme which assigns isolated learning objectives to each modality without mutual inhibition, and guarantee the cross-modal interaction simultaneously.

### 2.2 Contrastive Learning in MML

Contrastive learning (CL) [4] aims to learn an embedding space where positive samples are clustered together while negative samples are pushed apart. Traditionally, CL has been applied to unimodal scenarios, e.g., self-supervised learning [12, 16], domain generalization [18, 40] and few-shot learning [21, 38]. In recent years, multimodal contrastive representation learning (MCRL) [19, 24] has been proposed to learn a shared feature space where the semantically aligned cross-modal representations are acquired. In MCRL, the paired multimodal samples are viewed as positive samples while the mismatched sample pairs are considered as negative samples. The cross-modal contrastive loss aims to pull the positive representations close in the instance level. MCRL has achieved great success yet. Multimodal pretrained models [10] emerged based on it, e.g., the vision-language models UniCL [37], FILIP [39], audio-text model CLAP [7] and audio-visual model CAV-MAE [11]. However, these methods are designed to align shared information in different modalities while overlooking the learning about the modality-specific and complementary features. In this paper, we aim to achieve cross-modal interaction during the unimodal learning process via the complementary knowledge transfer based on CL.

## 3 METHODOLOGY

In this section, we analyze the modality competition problem and elaborate on the details of our proposed DI-MML. We mainly focus on a multi-class classification task with multimodal data.

### 3.1 Modality Competition Analysis

Let $x$ be a data sample and $y = [K]$ be the corresponding label. Without loss of generality, we consider two input modalities $x = [x^1, x^2]$.

Table 1: The modality competition analysis on CREMA-D, AVE and UCF101. The metric is the top-1 accuracy (%). 'Audio', 'Visual', 'Flow' and 'Image' denote the corresponding uni-modal performance in each dataset. 'Multi' is the multimodal performance. 'Uni1' and 'Uni2' mean unimodal training based on audio and visual data respectively for CREMA-D and AVE, while flow and image respectively for UCF101.

| Dataset | CREMA-D [3] | | | AVE [28] | | | UCF101 [26] | | |
|---|---|---|---|---|---|---|---|---|---|
| Method | Audio | Visual | Multi | Audio | Visual | Multi | Flow | Image | Multi |
| Uni1 | 65.59 | - | - | **66.42** | - | - | 55.09 | - | - |
| Uni2 | - | 78.49 | - | - | 46.02 | - | - | 42.96 | - |
| Joint training | 61.96 | 38.58 | 70.83 | 63.93 | 24.63 | 69.65 | 33.78 | 37.54 | 51.92 |
| MM Clf | 65.59 | 78.49 | 78.09 | 66.42 | 46.02 | 72.39 | 55.09 | 42.96 | 60.67 |
| Preds Avg | 65.59 | 78.49 | 82.66 | 66.42 | 46.02 | 69.40 | 55.09 | 42.96 | 64.43 |
| CM Dist | 63.17 | 77.28 | 82.93 | 62.94 | 41.79 | 67.41 | 54.30 | 42.93 | 64.45 |
| Ours | **66.67** | **78.90** | **83.74** | 64.18 | **49.25** | **75.37** | **58.52** | **48.59** | **65.79** |

In MML, we generally use two encoders $\phi^1, \phi^2$ to extract features of each modality: $\boldsymbol{h}^1 = \phi^1\left(\theta^1, \boldsymbol{x}^1\right)$ and $\boldsymbol{h}^2 = \phi^2\left(\theta^2, \boldsymbol{x}^2\right)$, where $\theta^1$ and $\theta^2$ are the parameters of encoders. And then, a fusion module is employed to integrate the information from two modalities and make predictions, i.e. $\psi\left(\boldsymbol{h}^1, \boldsymbol{h}^2\right)$, where $\psi$ denotes the fusion and prediction function. The overall function of multimodal model can be written as $f(x) = \psi\left(\phi^1\left(x^1\right), \phi^2\left(x^2\right)\right)$. Therefore, the cross-entropy loss for multimodal classification is:

$$\mathcal{L}_{CE}(x) = -\log \frac{\exp\left(f(x)_y\right)}{\sum_{k=1}^{K} \exp\left(f(x)_k\right)} \quad (1)$$

This is a uniform learning objective for both modalities. MML is expected to exploit the complementary information of all modalities to outperform unimodal learning, but the modality competition phenomenon limits the performance improvement of MML since the dominant modality will inhibit the learning process of other modalities. As demonstrated in Table 1, the unimodal performance from the traditional multimodal joint training severely underperforms the results from corresponding unimodal training. In particular, one of the two modalities could be severely suppressed, e.g., visual modality in CREMA-D and AVE, and the flow in UCF101.

Although several methods [22, 27, 30] have been proposed to alleviate the modality competition, we find that the culprit behind, a uniform learning objective for both modalities, has not been resolved. According to the loss function Eq. 1, we can obtain the gradient of the softmax logits output with ground-truth label $y$:

$$\frac{\partial \mathcal{L}_{CE}}{\partial f(x)_y} = \frac{\exp\left(f(x)_y\right)}{\sum_{k=1}^{K} \exp\left(f(x)_k\right)} - 1 \quad (2)$$

which is the gap between the predictive probability on ground truth with the value 1. If one modality performs better (i.e., the needed gradient strength should be low) and dominates the fusion feature, the strength of generated gradient with the uniform learning objective could be weak, which cannot satisfy the requirement of greater gradient strength for the weak modality, as illustrated in Figure 2. Therefore, *removing the uniform learning objective for encoder training* is the key to eliminating modality competition.

Intuitively, we can perform the detached unimodal learning for each encoder independently and then fuse their outputs (features or logits). As shown in Table 1, we fix the pretrained unimodal learned networks and fuse their information in two ways: (1) MM Clf, train a multimodal linear classifier with the output features; (2) Preds Avg, average the prediction of each modality. It is clear that they can achieve impressive improvement compared with joint training despite the restricted cross-modal interactions, indicating the necessity to eliminate competition in MML. However, there still remain some challenges. Firstly, due to the heterogeneity between modalities, independent unimodal training may lead to disparate latent feature spaces. The correlations between modalities are ignored, making it difficult to fuse information effectively. For example, MM Clf on CREMA-D and UCF101 is worse than Preds Avg since the heterogeneous feature spaces hinder the feature fusion. Secondly, according to [6], the cross-modal interactions in joint training can help to explore the complementary information that is hard to be learned with unimodal training. Independent encoder training blocks cross-modal interactions, thus, limiting the use of multimodal complementary knowledge. Here we apply naive cross-modal logit distillation in independently unimodal training, namely CM Dist, to achieve inter-modal knowledge transfer, enabling the multimodal interactions via prediction with multimodal data as in joint training. It can be seen that CM Dist is better than MM Clf and Preds Avg on CREMA-D and UCF101, showing the potential of cross-modal knowledge transfer for multimodal interactions. Nonetheless, the naive distillation does not consider the heterogeneity between the modalities so it does not work well always (perform worse on AVE), which motivates us to design more delicate cross-modal interactive behavior.

We then present our method in next subsection, which not only solves all of the above challenges but achieves consistent improvement for various datasets on both multi- and uni-modal accuracy.

## 3.2 Detached and Interactive MML

According to the above discussion, we separately train each modality's encoder to avoid modality competition. Meanwhile, we enable cross-modal interactions during the encoder training and fusion process, as well as inference, to exploit the complementary information between different modalities. The details are given below and the overall framework is shown in Figure 3.

**Detached unimodal training.** The network of each modality is updated only according to its own data and learning objectives, and there is no fusion during the update of encoders. Encoders $\phi^1, \phi^2$ are equipped with corresponding classifiers $\psi^1$ and $\psi^2$. Therefore, the logit output of modality $i$ is $z^i = f^i\left(x^i\right) = \psi^i\left(\phi^i\left(x^i\right)\right), i \in \{1, 2\}$.

**Figure 3: Overall framework of DI-MML. The encoders of each modality are trained with isolated learning objectives. The connections and interactions between modalities during encoder training are enabled by shared classifier and DUC loss.**

The classification loss $\mathcal{L}_{CE}^i (x^i)$ of each modality is independent with each other, exploiting informative knowledge for classification. **Interaction during encoder training.** To address the disparate feature spaces, we use a shared linear classifier (S-Clf) for different modalities to regulate the consistent feature space. Given the extracted features $h^i$, the logit output through the shared classifier is $sz^i = Wh^i + b$, where $W = [W_1, \cdots, W_K] \in R^{d \times K}$, $b \in R^d$ are the parameters of S-Clf and $d$ is the feature dimension. According to [20, 25], the paired features $h^1, h^2$ with label $y$ are optimized to maximize the similarity between them with the $y$-th vector $W_y$, and hence, S-Clf forces two modalities to locate at the same feature space using $W_y$ as the anchor. The corresponding loss for each modality is denoted as $\mathcal{L}_{CE}^{Si} (x^i)$.

Then, we need to enable the cross-modal interaction to exploit the complementary information. According to the analysis in Section 3.1, cross-modal knowledge transfer is a promising way for interactions. Considering the gap between modalities [36], we intend to transfer the modality-level complementarities for efficient knowledge transfer and importantly do not interfere with the learning of unimodal knowledge. To achieve this, we propose a novel Dimension-decoupled Unidirectional Contrastive (DUC) loss. Due to factors such as over-parameterization and implicit regularization [2, 42], deep networks tend to learn low-rank and redundant features, which motivates us to **compensate the ineffective information present in features with the effective cross-modal complementary information**.

First, we need to perform dimension separation to specify the effective and ineffective dimensions for each modality. We define the effective dimensions as dimensions with better discriminative knowledge. Therefore, we devise the dimension-wise prediction to evaluate the discrimination for each modality. With all the features from modality $i$, we can obtain the feature centroid of each class as:

$$\bar{h}_k^i = \frac{1}{N_k} \sum_{j=1}^N \mathcal{I} \{y_j = k\} h_j^i, \ \bar{h}_k^i = \left[\bar{h}_{k,1}^i, \bar{h}_{k,2}^i, ..., \bar{h}_{k,d}^i\right]^T \quad (3)$$

where $N$ is the number of all samples and $N_k$ is the number of samples belong to $k$-th class. And then, we can make dimension-wise evaluation by comparing the distance for each dimension with its dimensional centroid:

$$r_m^i = \frac{1}{N} \sum_{j=1}^N \mathcal{I} \left\{\arg\min_k d\left(h_{j,m}^i, \bar{h}_{k,m}^i\right) = y_j\right\}, m \in [d] \quad (4)$$

$d(\cdot, \cdot)$ is the distance function (Euclidean distance here). $r_m^i$ can be used to assess the effectiveness of dimension $m$ of modality $i$. Larger value indicates higher effectiveness on classification. Hence, the dimension separation principle is that the effective dimensions are represented with dimensions whose dimension-wise evaluation is greater than the mean value:

$$\begin{cases} r_m^i > \bar{r}^i & m \ is \ effective \\ r_m^i < \bar{r}^i & m \ is \ ineffective \end{cases} \quad (5)$$

where $\bar{r}^i = \frac{1}{d} \sum_{m=1}^d r_m^i$. Through this way, the feature dimensions of each modality are divided into effective group $d_e^i = \{m | r_m^i > \bar{r}^i\}$ and ineffective group $d_{ne}^i = \{m | r_m^i < \bar{r}^i\}$. The dimension separation is operated after some warmup epochs, see details in Algorithm 1 in Appendix.

Due to the heterogeneity between modalities, they do not shared all the effective dimensions. Hence, we then propose to transfer the effective information in modality 1 to the corresponding ineffective dimensions in modality 2 and vice verse, as shown in Figure 3. The knowledge transfer is performed by our proposed DUC loss:

$$\mathcal{L}_{DUC}^1 = \mathbb{E}_{(x_i^1, x_i^2)} \left[-\log \frac{\exp\left(-d\left(\tilde{h}_i^1, \tilde{h}_i^2\right)/T\right)}{\sum_j \exp\left(-d\left(\tilde{h}_i^1, \tilde{h}_j^2\right)/T\right)}\right]$$

$$\mathcal{L}_{DUC}^2 = \mathbb{E}_{(x_i^1, x_i^2)} \left[-\log \frac{\exp\left(-d\left(\hat{h}_i^1, \hat{h}_i^2\right)/T\right)}{\sum_j \exp\left(-d\left(\hat{h}_j^1, \hat{h}_i^2\right)/T\right)}\right] \quad (6)$$

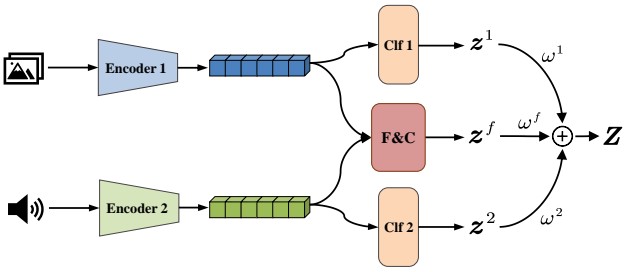

**Figure 4: During inference, the logit weighting is utilized on instance level.**

where $\tilde{h}_i^1 = \left[ h_{i,m}^1 | m \in d_{ne}^1 \cap d_e^2 \right]$, $\tilde{h}_i^2 = \left[ h_{i,m}^2 | m \in d_{ne}^1 \cap d_e^2 \right]$, $\hat{h}_i^1 = \left[ h_{i,m}^1 | m \in d_e^1 \cap d_{ne}^2 \right]$ and $\hat{h}_i^2 = \left[ h_{i,m}^2 | m \in d_e^1 \cap d_{ne}^2 \right]$. $T$ is the temperature. Notably, **the features of $\tilde{h}_i^2$ and $\hat{h}_i^1$ do not pass gradient backward**, which means we only allow the ineffective dimensions of modality 1 (2) to learn toward the corresponding effective dimensions of modality 2 (1), and do not update the effective dimensions of modality 2 (1) with DUC to prevent damage on the unimodal learning process. Hence, we let the complementary knowledge between modalities transfer unidirectionally and use the integrated knowledge for prediction to enable cross-modal interaction.

The final loss for modality $i$ can be calculated as:

$$\mathcal{L}^i = \mathcal{L}_{CE}^i + \lambda_s \mathcal{L}_{CE}^{Si} + \lambda_D \mathcal{L}_{DUC}^i \tag{7}$$

**Interaction during co-prediction.** The above training process does not directly utilize the multimodal data for completing tasks, therefore, in this stage we enable the interaction during the co-prediction process via training a fusion module with multimodal objective Eq. 1 while fixing the learned encoders.

### 3.3 Instance-level Weighting

In the training stage, we exploit the modality-level complementary information through DUC loss. However, the complementary capacities of the different modalities may also vary in different sample pairs [31]. Therefore, we propose a certainty-aware logit weighting strategy during inference to utilize the instance-level complementarities comprehensively, as demonstrated in Figure 4. We use the absolute certainty to evaluate the $j$-th instance reliability for each modality and their fusion:

$$c_j^i = \max_k softmax \left( z_j^i \right)_k, \ i \in \{1, 2, f\}, \ k \in [K]. \tag{8}$$

superscript $f$ denotes the output of fusion module. Then, the final output is:

$$Z_j = w_j^1 z_j^1 + w_j^f z_j^f + w_j^2 z_j^2$$

$$w_j^i = \frac{\exp \left( c_j^i / T \right)}{\exp \left( c_j^1 / T \right) + \exp \left( c_j^f / T \right) + \exp \left( c_j^2 / T \right)} \tag{9}$$

where more reliable modalities are assigned with higher weights.

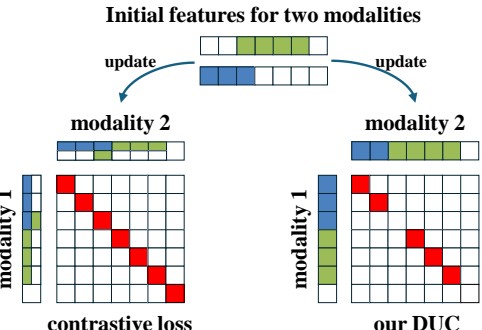

**Figure 5: Traditional contrastive loss is hard, aligning all the dimensions bidirectionally. Our DUC loss is soft, performing on part of dimensions and only transferring complementarities. Blue and green colors denote effective dimensions and white means ineffective dimension. Red color represents alignment between corresponding dimensions.**

### 3.4 Comparison with MCRL Loss

Previous multimodal contrastive loss [24] pays attention to searching for the semantic alignment between modalities, hence, the learning strength is bidirectional on the whole dimensions, i.e. the positive samples of two modalities move toward each other. Nevertheless, the alignment objective is too 'hard' that may lead to information loss, since there may be noise in part of the dimensions for specific modalities and complete alignment would partially preserve the noise, as illustrated in Figure 5. In contrast, our DUC loss is not intended to perform semantic alignment, but rather cross-modal transfer of complementary knowledge. Therefore, we decouple the feature dimensions and perform a unidirectional cross-modal knowledge transfer to enhance the dimensions with less informative knowledge while retaining effective information unique to the current modality. It can be seen that our DUC is more 'soft', and the dimensions in $d_e^1 \cap d_e^2$ are not required to align with each other, preserving the specific characteristics of each modality.

## 4 EXPERIMENTS

### 4.1 Dataset

We use four different multimodal datasets, *i.e.*, CREMA-D [3], AVE [28], UCF101 [26], and ModelNet40. CREMA-D is an audio-visual dataset for researching emotion recognition, comprising facial and vocal emotional expressions. Emotions are categorized into 6 types: happy, sad, angry, fear, disgust, and neutral. The dataset consists of 7442 segments, randomly divided into 6698 samples for training and 744 samples for testing. AVE is an audio-visual video dataset designed for audio-visual event localization, encompassing 28 event classes and 4,143 10-second videos. It includes both auditory and visual tracks along with secondary annotations. All videos are collected from YouTube. In our experiments, we extract frames from event-localized video segments and capture audio clips within the same segment, constructing a labeled multimodal classification dataset as in [8]. UCF101 is a dataset for action recognition comprising real action videos with 101 action categories, collected

**Table 2: Comparative analysis of different methods on CREMA-D, AVE, UCF101 and ModelNet40. The metric is the top-1 accuracy (%). 'Audio', 'Visual', 'Flow', 'Image', 'Front' and 'Rear' denote the corresponding uni-modal performance in each dataset. 'Multi' is the multimodal performance. 'Uni1' and 'Uni2' mean unimodal training based on audio and visual data respectively for CREMA-D and AVE, while flow and image for UCF101, front-view and rear-view for ModelNet40. The best is in bold, and the second best is underlined.**

| Dataset | CREMA-D [3] | | | AVE [28] | | | UCF101 [26] | | | ModelNet40 | | |
|---|---|---|---|---|---|---|---|---|---|---|---|---|
| Method | Audio | Visual | Multi | Audio | Visual | Multi | Flow | Image | Multi | Front | Rear | Multi |
| Uni1 | 65.59 | - | - | **66.42** | - | - | 55.09 | - | - | 89.63 | - | - |
| Uni2 | - | 78.49 | - | - | 46.02 | - | - | 42.96 | - | - | 88.70 | - |
| Joint training | 61.96 | 38.58 | 70.83 | 63.93 | 24.63 | 69.65 | 33.78 | 37.54 | 51.92 | 85.98 | 81.81 | 89.63 |
| MSES [9] | 62.50 | 37.90 | 70.43 | 63.93 | 24.63 | 69.65 | 33.99 | 37.19 | 51.76 | 85.98 | 81.81 | 89.63 |
| MSLR [41] | 63.04 | 41.13 | 71.51 | 61.19 | 24.63 | 68.91 | 33.44 | 37.77 | 52.60 | 86.22 | 82.17 | 89.59 |
| OGM-GE [22] | 61.29 | 39.27 | 71.14 | 62.45 | 27.39 | 69.12 | 40.73 | 33.44 | 53.56 | 86.35 | 82.09 | 89.30 |
| PMR [8] | 63.04 | 71.24 | 75.54 | 63.18 | 35.57 | 70.89 | 45.86 | 39.49 | 51.73 | 87.28 | 86.02 | 90.19 |
| UMT [6] | 65.46 | 75.94 | 77.42 | 65.42 | 42.29 | 73.88 | 55.41 | 45.15 | 61.51 | 88.33 | 87.76 | 90.80 |
| MM Clf | 65.59 | 78.49 | 78.09 | 66.42 | 46.02 | 72.39 | 55.09 | 42.96 | 60.67 | 89.63 | 88.70 | 90.19 |
| Preds Avg | 65.59 | 78.49 | 82.66 | 66.42 | 46.02 | 69.40 | 55.09 | 42.96 | 64.43 | 89.63 | 88.70 | 90.92 |
| Ours | **66.67** | **78.90** | **83.74** | 64.18 | **49.25** | **75.37** | **58.52** | **48.59** | **65.79** | **89.83** | **88.74** | **90.92** |

from YouTube. We treat the optical flow and images of the videos as two separate modalities. The dataset consists of 13,320 videos, with 9,537 used for training and 3,783 for testing. ModelNet40 is one of the Princeton ModelNet datasets [34] with 3D objects of 40 categories, consisting of 9,843 training samples and 2,468 testing samples. Following [33], we treat the front view and the rear view as two modalities in our experiments.

## 4.2 Experimental Settings

For the above four datasets, we used ResNet18 [13] as the backbone encoder network, mapping input data into 512-dimensional vectors. For the input data, for the CREMA-D and AVE datasets, audio modality data was transformed into spectrograms of size 257×1,004, and visual modality data consisted of 3(4 frames for AVE) randomly selected frames from 10-frame video clips, with image size of 224×224. For the UCF101 dataset, we randomly sampled contiguous 10-frame segments from videos during training, while testing, we sampled 10-frame segments from the middle of the videos. Optical flow modality data was of size 20×224×224, and visual modality data consisted of randomly sampled 1 frame. For the ModelNet40 dataset, we utilized front and back views as two modalities. For all visual modalities, we applied random cropping and random horizontal flipping as data augmentation during training; we resized images to 224×224 without any augmentation during testing. We trained all models with a batch size of 16, using SGD optimizer with momentum of 0.9 and weight decay of 1e-4, for a total of 150 epochs, with initial learning rate of 1e-3 decaying to 1e-4 after 70 epochs. For the training of fusion module, we trained for 20 epochs, with initial learning rate of 1e-3 decaying to 1e-4 after 10 epochs. All experiments were conducted on an NVIDIA GeForce RTX 3090 GPU and a 3.9-GHZ Intel Core i9-12900K CPU.

## 4.3 The Effectiveness of DI-MML

We compare DI-MML with various baselines and analyze the DUC loss to validate the effectiveness of our method.

**Table 3: The ablation study on CREMA-D and AVE.**

| TS | S-Clf | DUC | LW | CREMA-D | | | AVE | | |
|---|---|---|---|---|---|---|---|---|---|
| | | | | Audio | Visual | Multi | Audio | Visual | Multi |
| | | | | 61.96 | 38.58 | 70.83 | 63.93 | 24.63 | 69.65 |
| ✓ | | | | 65.59 | 78.49 | 78.09 | **66.42** | 46.02 | 72.39 |
| ✓ | ✓ | | | 66.26 | 79.70 | 79.70 | 64.43 | 44.78 | 72.14 |
| ✓ | ✓ | ✓ | | 66.67 | 78.90 | 82.80 | 64.18 | 49.25 | 72.89 |
| ✓ | ✓ | ✓ | ✓ | **66.67** | **78.90** | **83.74** | 64.18 | **49.25** | **75.37** |

**Comparison with other baselines.** The compared methods are divided into two groups: with and without the uniform objective for encoder training. Only the MM Clf, Preds Avg and our DI-MML do not utilize the uniform objective. The results are shown in Table 2, we not only report the multimodal performance and also the unimodal accuracy. To ensure the fairness of the comparison, we fix the parameters of their unimodal encoder networks after multimodal training, and evaluate their unimodal performance by training a classifier independently. It can be that the methods with the uniform objective (joint training, MSES, MSLR, OGM-GE, PMR and UMT) are all suffered from severe modality competition as their unimodal performance is generally lower than the best unimodal training counterpart, especially on Visual in CREMA-D and AVE, Flow in UCF101 and Rear in ModelNet40. MSES, MSLR, OGM-GE and PMR regulate the learning progress of modalities by adjusting the learning rates or gradients of different modalities, which alleviates modality competition to some extent, but they are difficult to completely eradicate it. UMT maintains the unimodal performance better, but it requires pretrained unimodal models for distillation, which is expensive and impractical. In contrast, our method completely avoid the modality competition, resulting in comparable or even the best unimodal performance (improved by up to 3.11% and 3.44% on Flow and Image of UCF101) and the best multimodal performance (improved by up to 6.32% on CREMA-D) on all four datasets. Besides, we do not require additional computational cost for encoder training. Compared with MM Clf and Preds

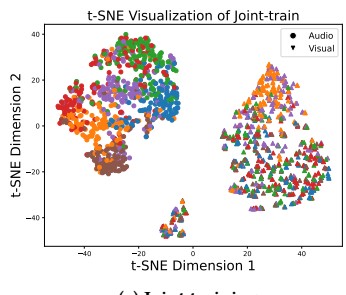

(a) Joint training

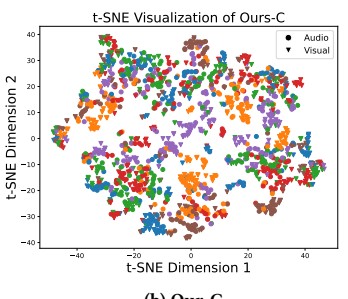

(b) Our-C

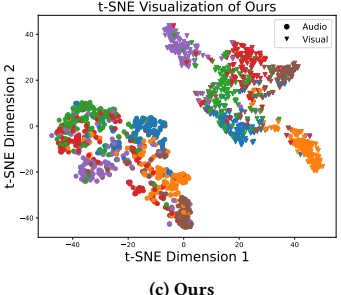

(c) Ours

Figure 6: The t-SNE feature visualization of each modality on CREMA-D. Different colors denote different classes.

Table 4: The performance comparison with various contrastive losses. 'A' and 'V' denote Audio and Visual.

| Dataset | CREMA-D | | | AVE | | |
|---------|---------|---------|-------|---------|---------|-------|
| Method | A | V | Multi | A | V | Multi |
| w/o DUC | 66.26 | 79.70 | 82.47 | **64.43** | 44.78 | 73.13 |
| Our-C | 65.73 | 79.17 | 81.72 | 63.18 | 46.77 | 71.39 |
| Our-DBC | 65.99 | **79.84** | 82.12 | 63.18 | **49.50** | 73.13 |
| Ours | **66.67** | 78.90 | **83.74** | 64.18 | 49.25 | **75.37** |

Table 5: The number of effective dimensions for each modality on three datasets. 'Overlap' denotes $\left| d_e^1 \cap d_e^2 \right|$. The results are obtained from the model after warmup epochs.

| | CREMA-D | AVE | UCF101 |
|---------|---------|-----|--------|
| Audio/Flow eff | 259 | 258 | 246 |
| Visual/Image eff | 262 | 291 | 249 |
| Overlap | 156 | 142 | 138 |

Avg, our DI-MML enables cross-modal interactions and complementary knowledge transfer during the encoder training. Therefore, our method can achieve both better multimodal and unimodal performance on these datasets. These results show that our approach is indeed competition-free, which is the key difference compared with previous methods. It also suggests that the proposed cross-modal interactions via knowledge transfer are effective.

**Ablation study.** There are four main components in our method: two-stage training scheme (TS, i.e. encoders and fusion module are trained separately), shared classifier (S-Clf), dimension-decoupled unidirectional contrastive loss (DUC), and logit weighting (LW). Here, we perform an ablation study to explore the influence of various combinations of these components. The experiments are conducted on CREMA-D and AVE. As demonstrated in Table 3, applying TS denotes the MM Clf method, which is better than Joint training because there is no modality competition. The shared classifier can align a feature space for different modalities and achieve considerable improvement on CREMA-D. The DUC loss facilitates cross-modal interaction and knowledge transfer, helping to achieve complementary knowledge utilisation at the modality level. Similarly, LW enables complementary knowledge integration at the instance level, both of them are important for multimodal performance enhancement. As discussed above, the four components are all essential in our method.

**Analysis on DUC loss.** The DUC loss is the central technique in our method to enhance the cross-modal interaction during the encoder training stage. In Section 3.4, we compare the differences between DUC and traditional multimodal contrastive learning loss in terms of aim and formality. Here, we give more experimental results to show the superiority of our method. The results are shown

in Table 4, where '-C' denotes replacing our DUC loss with traditional multimodal contrastive loss while '-DBC' means dimension-decoupled bidirectional contrastive loss, i.e., $\tilde{h}_i^2$ and $\hat{h}_i^1$ can pass gradients backward in Eq. 6, suggesting that $\tilde{h}_i^1$ and $\tilde{h}_i^2$ ($\hat{h}_i^1$ and $\hat{h}_i^2$) move toward each other as traditional contrastive loss. It is clear that using traditional contrastive loss performs worst as it does not consider retaining the modality-wise complementary information. Applying DBC achieves improvement since the it does not affect the learning of effective dimensions shared by modalities (i.e., $d_e^1 \cap d_e^2$). However, the noise information in the ineffective dimensions is preserved as illustrated in Figure 5. Our DUC loss both preserves the unimodal knowledge and facilitates inter-modal cooperation through complementary knowledge transfer, resulting in the best multimodal results. In Figure 6, we demonstrate the t-SNE [29] feature visualization for each modality on CREMA-D. Figure 6a showcases that there are no clear decision boundaries for visual features for joint training, consistent with its poor performance. As shown in Figure 6b, although applying contrastive loss in our method compensates for the gap between different modalities in feature space, the noise in visual modality is also preserved and transferred to audio modality to some extent, leading to worse multimodal performance. With the optimization of our DUC loss as shown in Figure 6c, the features of both modalities are more clearly clustered, besides, share a more similar distributional structure.

**Analysis on dimension separation.** In this paper, we perform the dimension separation to divide dimensions into effective and ineffective parts. The separation results are displayed in Table 5. The effective dimensions for both modalities take up about half or more (feature is a 512-dimensional vector), and their overlap also accounts for only about half of effective dimensions, indicating that there are enough dimensions to ensure cross-modal knowledge transfer.

Table 6: The performance of effective and ineffective dimensions of each modality.

| Dataset | CREMA-D | | | AVE | | |
|---|---|---|---|---|---|---|
| Modality | all | eff | ineff | all | eff | ineff |
| Audio | 58.60 | 54.71 | 31.59 | 59.70 | 50.25 | 43.03 |
| Visual | 46.37 | 31.99 | 23.79 | 25.12 | 21.64 | 18.91 |

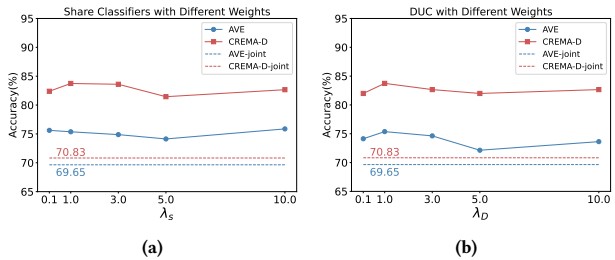

(a)             (b)

Figure 7: Comparison with different values of $\lambda_s$ and $\lambda_D$ on CREMA-D. Our method is robust to two hyperparameters.

The performance of corresponding dimension sets of effectiveness and ineffectiveness is shown in Table 6. When we evaluate the performance of effective dimension set, the values of ineffective dimensions are set to 0 and vice verse, removing its influence on the output prediction. The performance of effective dimensions is much better than that of ineffective dimensions, indicating that our dimension separation scheme is reasonable and effective.

## 4.4 Robustness Validation

**Effective dimension evaluation.** In this paper, we devise the dimension-wise prediction as in Eq. 4 to evaluate the effectiveness of each dimension. Here, we compare our dimension-wise prediction with two other evaluation metrics: L2-norm and Shapley Value. According to [23], the L2-norm of the features gives an indication of their information content, thus it can be used as a metric to measure the effectiveness of each dimension. And shapley value can also be used to identify important features (dimensions here) by removing specific content for prediction. As depicted in Table 7, our proposed framework has significant enhancements with any evaluation method, showing the robustness of our DI-MML framework. Besides, among the three methods, our dimension-wise prediction performs the best on different datasets, indicating its validity for evaluating the dimensionally discriminative information.

**Hyperparameter sensitivity.** In the calibration of our DI-MML, we encounter two hyperparameters to determine: $\lambda_s$ and $\lambda_D$ in Eq. 7, determining the strength for feature space alignment and cross-modal knowledge transfer respectively. Here, we explore the effects of them by varying their values as illustrated in Figure 7. It is clear that the performance on DI-MML is marginally affected by $\lambda_s$ and $\lambda_D$, suggesting the insensitivity of our method to hyperparameters. Despite some fluctuations in performance with hyperparameters, our method still demonstrates excellent effectiveness, i.e., being consistently better than joint training. In this paper, we select $\lambda_s = 1$ and $\lambda_D = 1$ for the best accuracy according to the obtained results.

Table 7: The performance of different methods for evaluating the effectiveness of each dimension.

| Dataset | CREMA-D | AVE |
|---|---|---|
| Method | Multi | Multi |
| Joint training | 70.83 | 69.65 |
| L2-norm | 83.60 | 73.17 |
| Shapley value | 81.58 | **75.37** |
| Dimension-wise prediction | **83.74** | **75.37** |

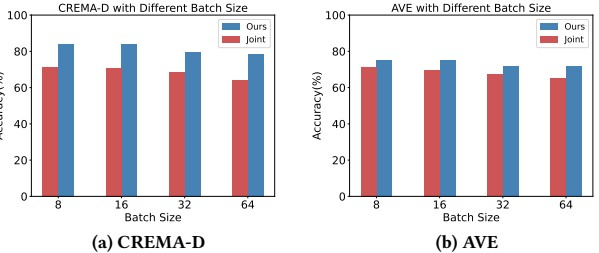

(a) CREMA-D          (b) AVE

Figure 8: Comparison results with different batch sizes. Our method consistently outperforms joint training.

**Robustness on Batch size.** To analyze the effect of batch size of our method, we demonstrate the results with different batch sizes on CREMA-D and AVE, varying from 8 to 64. It can be seen that small batch size could lead to better performance on both joint training and DI-MML, and our method consistently outperforms joint training on all the batch sizes, indicating the robustness of our DI-MML with respect to batch size. In this paper, we set batch size to 8 to get the best results for the four datasets.

## 5 CONCLUSION

In this paper, we analyze the multimodal joint training and argue that the modality competition problem comes from the uniform learning objective for different modalities. Therefore, we propose to train multimodel encoders separately to avoid modality competition. To facilitate the feature space alignment and cross-modal interaction, we devise a shared classifier and the dimension-decoupled unidirectional contrastive loss (DUC) to achieve modality-level complementarities utilization. And then, the learned encoders are frozen and a fusion module is updated for interaction during co-prediction. Considering the reliability differences on various sample pairs, we further propose the certainty-aware logit weighting strategy to exploit instance-level complementarities comprehensively. Through extensive experiments, our DI-MML outperforms all competing methods in four datasets. We also showcase that our method can further promote the unimodal performance instead of inhibiting them. In the future, we can investigate other types of cross-modal interactions and focus on multimodal tasks such as detection or generation instead of only classification. Besides, identifying the specific semantics in each dimension may be helpful to further evaluate the informative dimensions.

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
