# OpenReview forum: "Detached and Interactive Multimodal Learning"
_acmmm.org/ACMMM/2024/Conference — MM2024 Poster_

### Official Review · Reviewer_h8sZ · 2024-05-09

**Rating:** 4
**Confidence:** 4

**Summary:**

This paper proposes Detached and Interactive MultiModal Learning (DI-MML) to address the modality competition problem in multimodal learning. In contrast to existing works that adjust modality-specific learning strength, DI-MML focuses on cross-modal detachment and interaction in the way of separated modality training, shared classifier, and dimension-decoupled unidirectional contrastive loss. Empirical results on multimodal classification datasets demonstrate the performance of DI-MML.

**Strengths:**

1. The illustrated figures are clear and helpful for understanding.
2. The codebase attached in the supplementary material is well-organized.
3. The experimental results are relatively sufficient and verify the effectiveness of each module, such as DUC loss.

**Limitations:**

1. The related works are not comprehensive. To my knowledge, a classical method named Gradient-Blending [1] and a recent method named I2MCL [2] are not mentioned or involved in this paper, both of which are proposed to solve the same problem, modality competition, as this work. Especially, Gradient-Blending also considers detaching multimodal learning, so I highly recommend comparing it empirically.
2. According to Table 3, it is observed that TS module (training separately) contributes most to solving the problem of modality competition, so it is necessary to explore whether a suitable reweighting of $L_{CE}^{i}$ is enough to improve the performance, that is, $L_{CE}^1+L_{CE}^2$ should be $\lambda_{CE}^{1}L_{CE}^{1}+\lambda_{CE}^{2}L_{CE}^{2}$.
3. The explanation on Figure 6 is confusing. Since this method aims to interact among multiple modalities, there should not be a clear boundary between the two modalities in Figure 6(c). Instead, the phenomenon in Figure 6(b) is more consistent with the expectation, that is, audio and visual features interact with each other to prevent modality competition. I think the reason why there is a clear boundary in Figure 6(c) may be because the authors discard the overlap dimensions $d_e^1\cap d_e^2$ when calculating effective dimensions in DUC loss, so some of the dimensions are interacted while others not, thereby showing a clear boundary.
4. A question is how to solve the problem with three or more modalities with this DI-MML method?

[1] Wang, Weiyao, Du Tran, and Matt Feiszli. "What makes training multi-modal classification networks hard?." Proceedings of the IEEE/CVF conference on computer vision and pattern recognition. 2020.

[2] Zhou, Yuwei, et al. "Intra-and Inter-Modal Curriculum for Multimodal Learning." Proceedings of the 31st ACM International Conference on Multimedia. 2023.

**Suitability:**

3

---

### Official Review · Reviewer_73YU · 2024-05-24

**Rating:** 3
**Confidence:** 3

**Summary:**

This paper introduces DI-MML, a detached Multimodal Learning (MML) framework that aims to address the modality competition issue inherent in traditional joint learning approaches. DI-MML employs a detached training strategy for each modality-specific encoder to avoid competition. The paper demonstrates DI-MML's superior performance across various datasets, including audio-visual, flow-image, and front-rear view pairs.

**Strengths:**

1. This paper addresses a critical challenge in multimodal learning - modality competition. By introducing a novel competition-free training scheme for MML.

2. The authors propose a Detached Interactions MML (DI-MML) framework, which is an innovative method that combines detached training with cross-modal interactions, making full use of modality-level complementarities.

3. The paper proposes specific strategies, including a shared classifier, a Dimension-decoupled Unidirectional Contrastive (DUC) loss, and a certainty-aware logit weighting strategy. These strategies help tackle disparate feature spaces and the lack of cross-modal interactions.

**Limitations:**

1.The paper has repeatedly stated "completely avoid modality competition while ensuring complementary cross-modal interactions in MML". How does the author substantiate the claim of completely avoiding modality competition rather than mitigating it to some extent? In the AVE dataset, the performance of the Audio modality in your proposed method doesn't surpass the performance of independently trained Audio modality.

2.What is the rationale behind choosing the CREMA-D, AVE, UCF101 datasets instead of a specific multi-modal dataset related to a particular task? Is there a correlation between the datasets chosen? To draw more reliable conclusions, we recommend selecting larger datasets, such as the VGG Sound and Kinetics-400 datasets used by the comparative methods discussed in this paper.

3.Is your method adapted only for classification tasks? If so, this might pose constraints on the practical application of the proposed method in the paper.

4.In the original paper of the comparative methods in this paper, the accuracy of the UCF101's RGB encoder and Opt-Flow encoder were 77.08% and 74.99% respectively. However, the accuracy reported in your paper is only around 50%.

**Suitability:**

2

---

### Official Review · Reviewer_UFz1 · 2024-05-24

**Rating:** 4
**Confidence:** 3

**Summary:**

This paper focuses on the problem of multimodal learning. The authors propose a framework named DI-MML, which avoids modality competition by separately training each modality encoder with isolated learning objectives. It enhances cross-modal interaction via a shared classifier and uses a dimension-decoupled unidirectional contrastive (DUC) loss for knowledge transfer. A certainty-aware logit weighting strategy is employed to leverage complementary information during inference. Experiments on various datasets show the superior performance of DI-MML.

**Strengths:**

The experimental evaluations are conducted on multiple different and challenging multimodal datasets, and the proposed solution demonstrates superior performance compared to existing methods. Additionally, detailed ablation experiments are performed to verify the influence of network structure design and hyperparameter selection on the accuracy of the proposed method.

**Limitations:**

1) The authors have provided a dual-modal recognition scheme and experiments. Could the proposed method be extended to tasks involving three or more modalities, such as text-image-audio analysis?

2) The two modalities used have similar data and network structures. Can this method be extended to multimodal data with greater differences or different kind of network (2D ResNet and PointNet), such as text and images, images and point clouds, etc.?

3) The authors have studied the competitive relationships between modalities. However, when one modality is contaminated, such as noisy audio or blurred images, would the proposed method be affected?
4) Tables 1 and 2 use the UCF101 dataset, which has achieved approximately 99% classification accuracy. However, the accuracy presented in the tables differs significantly. Please provide some clarification on this discrepancy.

**Suitability:**

3

---

### Meta-Review · Area_Chair_hK5R · 2024-07-03

**Recommendation:** Accept (Poster)
**Confidence:** 5

**Metareview:**

The final ratings of this paper are three borderline accept. After rebuttal, all reviewers agree that the paper's strengths warrant acceptance, and the AC agrees.

---

### Meta-Review · Senior_Area_Chairs · 2024-07-10

**Recommendation:** Accept (Poster)
**Confidence:** 4

**Metareview:**

This paper received mixed ratings initially. After rebuttal, all the reviewers tend to accept the paper. SAC and AC agree with reviewers and recommend acceptance of the paper.